# Precision RNAi using synthetic shRNAmir target sites

**Thomas Hoffmann[1,2†‡], Alexandra Hörmann[3†], Maja Corcokovic[3†],
Jakub Zmajkovic[1], Matthias Hinterndorfer[1], Jasko Salkanovic[3], Fiona Spreitzer[3],
Anna Köferle[3], Katrin Gitschtaler[3], Alexandra Popa[3], Sarah Oberndorfer[3],
Florian Andersch[1], Markus Schaefer[1], Michaela Fellner[1], Nicole Budano[3],
Jan G Ruppert[3], Paolo Chetta[3], Melanie Wurm[3], Johannes Zuber[1,4*],
Ralph A Neumüller[3*]**

[1]Research Institute of Molecular Pathology (IMP), Campus-Vienna-Biocenter 1,
Vienna, Austria; [2]Vienna BioCenter PhD Program, Doctoral School of the University
at Vienna and Medical University of Vienna, Vienna BioCenter (VBC), Vienna, Austria;
[3]Boehringer Ingelheim RCV GmbH & Co KG, Doktor-Boehringer-Gasse, Vienna,
Austria; [4]Medical University of Vienna, Vienna BioCenter, Vienna, Austria

**\*For correspondence:**
johannes.zuber@imp.ac.at (JZ);
ralph.neumueller@boehringer-
ingelheim.com (RAN)

[†]These authors contributed
equally to this work

**Present address:** [‡]Advantage
Therapeutics GmbH, Karl-Farkas-
Gasse 22, Vienna, Austria

**Competing interest:** See page
12

**Reviewing Editor:** Michael B
Eisen, University of California,
Berkeley, United States

**Abstract** Loss-of-function genetic tools are widely applied for validating therapeutic targets, but their utility remains limited by incomplete on- and uncontrolled off-target effects. We describe artificial RNA interference (ARTi) based on synthetic, ultra-potent, off-target-free shRNAs that enable efficient and inducible suppression of any gene upon introduction of a synthetic target sequence into non-coding transcript regions. ARTi establishes a scalable loss-of-function tool with full control over on- and off-target effects.

## eLife assessment

This manuscript describes a **valuable** method to study the mechanism of action of essential genes and novel putative drug targets. Evidence for the effectiveness of the system, which is based on engineering pre-validated targets for RNA-mediated knockdown into genes of interest, is **compelling**, and the method should find use as an orthogonal method for generating gene-specific knockdowns.

## Introduction

Drug development is guided by genetic loss-of-function (LOF) experiments that validate a therapeutic target, study its general and disease-specific functions, and thereby model and benchmark expected activities of inhibitory molecules. Applied genetic tools include RNAi, CRISPR/Cas9, and site-specific recombination technologies such as the Cre-Lox or Flp-FRT systems (*Mohr et al., 2014*; *Housden et al., 2017*). While these technologies have undoubtedly revolutionized genetic screening, target identification and validation, each method is associated with drawbacks that limit the usability for certain aspects of target validation. Specifically, RNAi is prone to off-target effects (*Jackson et al., 2003*; *Scacheri et al., 2004*; *Lin et al., 2005*) and insufficient knockdown levels, while CRISPR/Cas9-based methods are associated with off-target effects and incomplete LOF across cell populations. These limitations are particularly problematic for candidate targets in oncology, which should ideally be validated genetically in cancer cell lines and tumor models in vivo. In both cases, insufficient LOF or off-target effects can lead to far-reaching misconceptions about target suppression effects. Outgrowth of wild-type clones that retain gene function upon CRISPR- and recombination-based gene editing

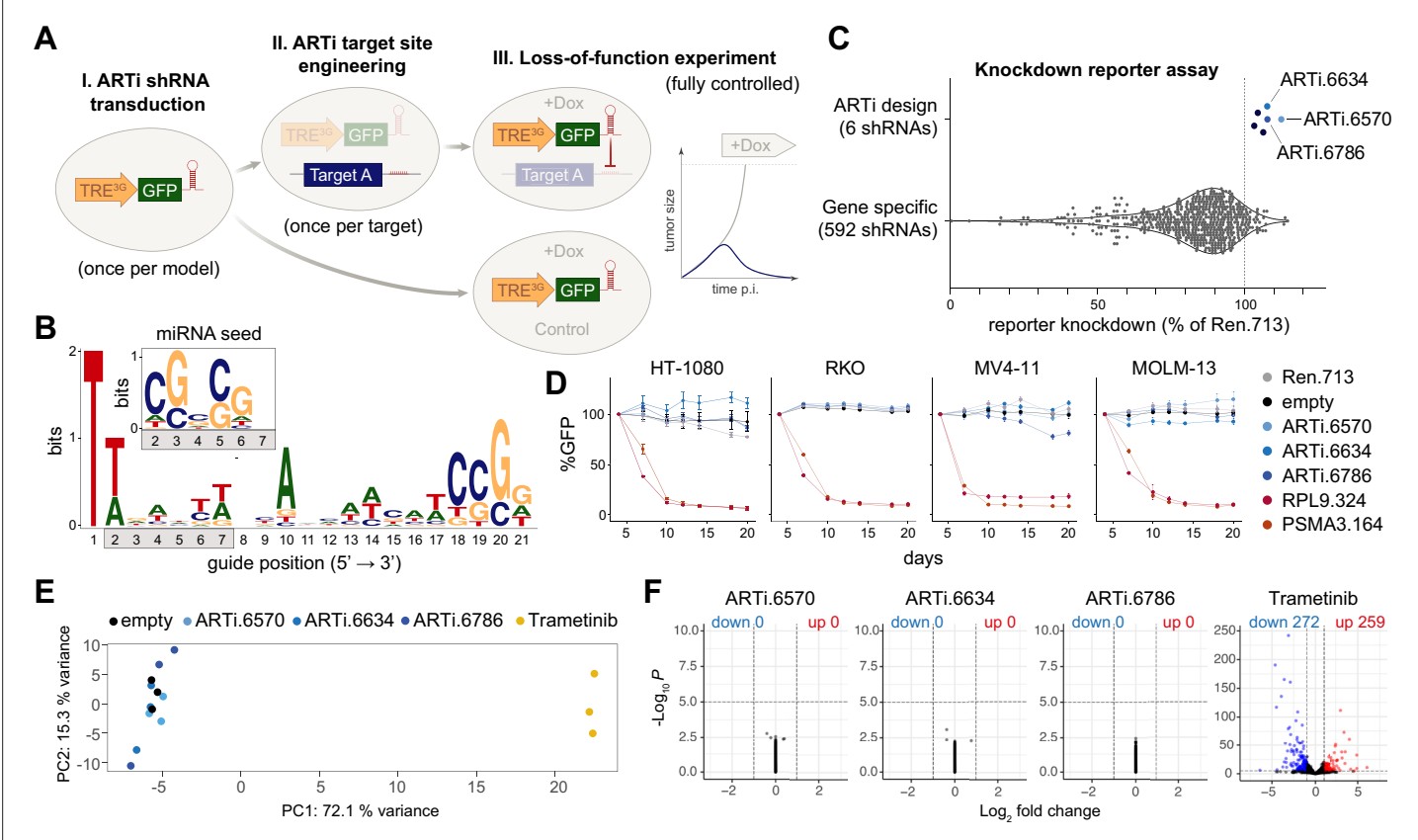

**Figure 1.** Design and selection ARTi-shRNAmirs. (**A**) Schematic outline of the ARTi approach. (**B**) Sequence logo (https://weblogo.berkeley.edu) displaying nucleotide position biases of 2161 shRNAs with exceptionally high Designer of Small Interfering RNA (DSIR) scores (>105). Inlay depicts miRNA seed sequence biases. (**C**) Reporter assay comparing gene-specific shRNAs to ARTi-shRNAmirs. (**D**) Competitive proliferation assays in human cell lines after transduction with ARTi-shRNAmirs, neutral (shRen.713) and essential control shRNAs (shRPL9.324 or shPSMA3.164) (n=3, error bars represent SD). (**E**) Principal component (PC) analysis of gene expression profiling upon stable expression of indicated shRNAs or treatment with MEK inhibitor (trametinib) in RKO cells. (**F**) Volcano plots visualizing de-regulated genes upon expression of indicated shRNAs and trametinib treatment in RKO cells compared to empty vector control. ARTi, artificial RNA interference; shRNAmirs, micro-RNA embedded shRNAs.

The online version of this article includes the following figure supplement(s) for figure 1:

**Figure supplement 1.** Design and selection ARTi-shRNAmirs.

---

can result in transient phenotypes that complicate data interpretation. Similarly, insufficient knockdown or uncontrolled off-target effects induced by RNAi can lead to an unjustified de-prioritization or pursuit of candidate targets, respectively. Prior to initiating the time- and resource-intense process of drug development, more informative target validation assays would be highly desirable.

## Results

To develop such an assay system, we reasoned that instead of using gene-specific LOF triggers for every new candidate gene, the expression of any gene could be efficiently suppressed by engineering the target site of a pre-validated, highly potent, synthetic short-hairpin RNA (shRNA) into its exonic sequence (**Figure 1A**). Besides ensuring efficient target knockdown in a highly standardized manner, such an approach would also provide control over off-target activities through expressing the shRNA side-by-side in target-site engineered and wildtype cells. This approach leaves the genome engineering procedure, needed to integrate the pre-validated artificial RNA interference (ARTi) target site into a gene of interest, as the only potential source of off-target effects. As suitable RNAi system, we chose optimized micro-RNA embedded shRNAs (shRNAmirs) in the miR-E backbone (**Fellmann et al., 2013**), which do not interfere with endogenous miRNA processing (**Premsrirut et al., 2011**) and can be expressed from tet-responsive elements and other Pol-II promoters in the

3'-UTR of fluorescent reporter genes, thus providing a versatile system for inducible RNAi (*Zuber et al., 2011*). To identify potent ARTi sequences with minimal off-target activity, we analyzed the nucleotide composition of shRNAs that reach exceptionally high-performance scores in common shRNA design algorithms (*Vert et al., 2006*) and of miRNA seed sequences with exceptionally low off-target scores according to siSPOTR (*Boudreau et al., 2013*). By merging nucleotide biases identified in both analyses, we derived a 22-nt base composition matrix for the design of ARTi shRNAmirs (TTCGWWWWNNAHHWWCATCCGGN; W = A/T, H = A/T/C; N = A/T/G/C) (*Figure 1B* and *Figure 1—figure supplement 1A and B*). To further reduce possible off-target effects, we eliminated all shRNAs whose extended seed sequence (guide positions 2–14) had a perfect match in the human or mouse transcriptome and, finally, selected six top-scoring ARTi predictions for experimental validation.

We tested these ARTi-shRNAmirs using an established knockdown reporter assay (*Fellmann et al., 2013*) for their ability to suppress expression of a GFP transgene that harbors the respective target sites in its 3'-UTR. In all six cases, ARTi-shRNAmir matched or outperformed previously validated highly potent shRNAmirs targeting Renilla luciferase or PTEN (*Fellmann et al., 2013*; *Figure 1—figure supplement 1C*). We compared these results to a panel of 592 gene-specific shRNAs that were tested using the same assay and found that ARTi-shRNAmirs ranked among top-performing shRNAmirs overall (*Figure 1C*). Next, we selected the three top-performing ARTi-shRNAmirs and evaluated possible off-target activities using competitive proliferation assays and transcriptome profiling. ARTi-shRNAmir expression had no effects on proliferation or survival in four human and three mouse cell lines (*Figure 1D* and *Figure 1—figure supplement 1D*), with the exception of ARTi.6634 and ARTi.6786, which induced a mild fitness defect in the murine leukemia cell line RN2 and MV4-11, respectively. In contrast to the effects of a MEK inhibitor trametinib, which we included as a positive control, stable expression of ARTi-shRNAmirs had only marginal effects on the transcriptome in two commonly used human cell lines (*Figure 1E and F* and *Figure 1—figure supplement 1E and F*) and no effect in proliferation assays (*Figure 1—figure supplement 1G*), indicating that they do not trigger major off-target effects, even in the absence of their respective target site. Together, these studies established a set of highly potent, off-target-free ARTi-shRNAmirs, among which we selected ARTi.6570 (ARTi-shRNAmir) for further investigations.

To establish ARTi as a method for target validation, we performed ARTi-based LOF experiments in cancer cell lines and xenograft models for three prominent oncology targets: EGFR and KRAS, which act as driving oncogenes in various cancer types (*Hynes and MacDonald, 2009*; *Punekar et al., 2022*), and STAG1, which has been identified as a synthetic lethal interaction with recurrent LOF mutations of STAG2 (*Bailey et al., 2021*; *van der Lelij et al., 2020*; *Benedetti et al., 2014*; *van der Lelij et al., 2017*). To establish an ARTi-repressible version of oncogenic EGFR[del19], we cloned ARTi-shRNAmir target sequences into an expression construct encoding EGFR[del19] fused to dsRed (*Figure 2A*), which rendered Ba/F3 cells cytokine-independent and sensitive to EGFR inhibition (*Figure 2—figure supplement 1A*). We introduced this construct into human EGFR[del19]-dependent PC-9 lung adenocarcinoma cells and subsequently knocked out the endogenous *EGFR* gene. The EGFR[del19]::V5::dsRed::ARTi transgene fully rescued the loss of endogenous EGFR[del19], while doxycycline (dox)-induced expression of the ARTi-shRNAmir (ARTi.6570) strongly inhibited proliferation of PC-9 cells and triggered a near-complete suppression of the EGFR[del19]::V5::dsRed::ARTi protein (*Figure 2B and C* and *Figure 2—figure supplement 1B*). Consistently, in RNA-sequencing we observed an almost complete drop of reads mapping to the codon-optimized *EGFR[del19]::V5::dsRed::ARTi* transgene upon dox-inducible expression of the ARTi-shRNAmir, as well as a downregulation of *DUSP6* and other canonical targets of RAF-MEK-ERK signaling, which acts as key effector pathway of EGFR[del19] (*Figure 2—figure supplement 1C-E*).

To evaluate whether ARTi can recapitulate drug activities in vivo, we xenotransplanted *EGFR[del19]::V5::dsRed::ARTi* engineered PC-9 cells harboring the dox-inducible ARTi-shRNAmir and treated recipient mice upon tumor formation with either dox or the clinically approved EGFR inhibitor osimertinib. Dox-induced expression of the ARTi-shRNAmir led to rapid and durable tumor regression that was indistinguishable from the effects of osimertinib (*Figure 2D*) and not observed in parental PC-9 cells that lack the ARTi target site (*Figure 2—figure supplement 1F*). We therefore conclude that ARTi-induced phenotypes are to be attributed to on-target effects that predict the activity of advanced small-molecule inhibitors in vivo.

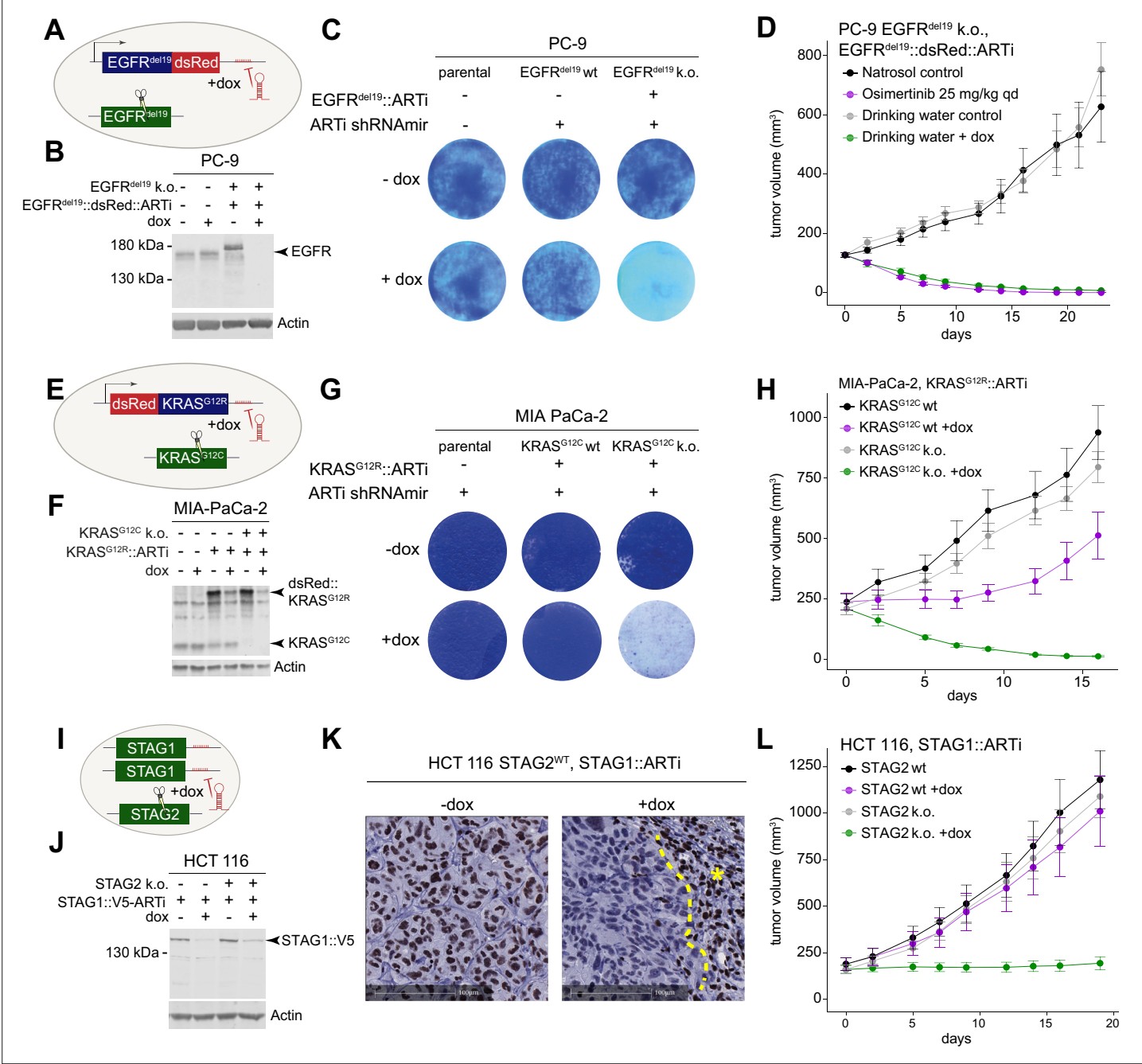

**Figure 2.** Experimental validation of the artificial RNA interference (ARTi) approach. (**A**) Schematic of EGFR[del19]::V5::dsRed::ARTi engineering in PC-9 cells. Blue color denotes overexpressed ARTi variant. Green denotes endogenous gene. (**B**) Western blot demonstrating knockdown of EGFR[del19]::V5::dsRed::ARTi. Western blot is a representative example of three independent biological repeat experiments. (**C**) Proliferation assay and crystal violet staining of parental and engineered PC-9 cells in the absence or presence of doxycycline (dox). Crystal violet staining is a representative example of two independent biological repeat experiments.( **D**) In vivo experiment comparing dox-induced EGFR[del19]::V5::dsRed::ARTi knockdown to pharmacological EGFR[del19] inhibition. Mean tumor volume and ± SEM is plotted for all in vivo experiments. (**E**) Schematic of MIA PaCa-2 engineering. Blue color denotes overexpressed ARTi variant. Green denotes endogenous gene. (**F**) Western blot for KRAS and Actin in indicated engineered MIA PaCa-2 cells in the presence and absence of dox. Western blot is a representative example of two independent biological repeat experiments. (**G**) Proliferation assay and crystal violet staining of parental and engineered MIA PaCa-2 cells in the absence or presence of dox. Crystal violet staining is a representative example of two independent biological repeat experiments. (**H**) Growth curves of tumors implanted with engineered MIA PaCa-2 cells in the absence and presence of dox in vivo. (**I**) Schematic of C-terminal endogenous tagging of STAG1. Green color denotes endogenous genes. (**J**) Western blot demonstrating knockdown of STAG1-ARTi. Western blot is a representative example of three independent biological repeat experiments. (**K**) Immunohistochemistry staining of STAG1 in engineered HCT 116 cells in the absence and presence of dox. Asterisk marks an area of

*Figure 2 continued on next page*

*Figure 2 continued*

murine fibroblasts that serve as an internal positive control. (**I**) Growth curves of tumors implanted with engineered HCT 116 cells in the absence and presence of dox.

The online version of this article includes the following source data and figure supplement(s) for figure 2:

**Source data 1.** Original blots for *Figure 2B* and *Figure 2—figure supplement 1B*.

**Source data 2.** Original blots for *Figure 2F*.

**Source data 3.** Original blots for *Figure 2J*.

**Figure supplement 1.** Validation of artificial RNA interference (ARTi) in vitro and in vivo – EGFR.

**Figure supplement 2.** Validation of artificial RNA interference (ARTi) in vitro and in vivo – KRAS.

**Figure supplement 2—source data 1.** Original blots for *Figure 2—figure supplement 2D*.

**Figure supplement 3.** Validation of artificial RNA interference (ARTi) in vitro and in vivo – STAG1.

**Figure supplement 3—source data 1.** Original blots for *Figure 2—figure supplement 3B*.

In a second study, we used ARTi to investigate KRAS$^{G12R}$, an oncogenic KRAS variant with no available in vivo compatible inhibitors. To establish a suitable KRAS$^{G12R}$-driven model, we engineered a *dsRed::KRAS$^{G12R}$-ARTi* transgene and the dox-inducible ARTi-shRNAmir into KRAS$^{G12C}$-dependent MIA PaCa-2 pancreatic adenocarcinoma cells and subsequently knocked out endogenous *KRAS* alleles (*Figure 2E* and *Figure 2—figure supplement 2A*). As expected, switching the driving oncogene from KRAS$^{G12C}$ to KRAS$^{G12R}$ rendered MIA PaCa-2 cells resistant to the KRAS$^{G12C}$ inhibitor AMG-510 (*Fakih et al., 2020*; *Lanman et al., 2020*; *Fakih et al., 2019*; *Figure 2—figure supplement 2B*). ARTi-shRNAmir induction led to a strong suppression of dsRed::KRAS$^{G12R}$ expression at the mRNA and protein level (*Figure 2F* and *Figure 2—figure supplement 2C and D*) and marked antiproliferative effects (*Figure 2G*). In xenograft experiments, ARTi-mediated suppression of KRAS$^{G12R}$ led to full tumor regression in the absence of KRAS$^{G12C}$ (*Figure 2H*), while tumors harboring both oncogenic KRAS alleles only displayed a delay in tumor progression, suggesting that both oncogenes contribute to tumor growth in vivo.

To evaluate STAG1 as synthetic-lethal dependency that is relevant in a wide range of STAG2-deficient cancers (*Figure 2—figure supplement 3A*; *Bailey et al., 2021*; *van der Lelij et al., 2020*; *Benedetti et al., 2014*; *van der Lelij et al., 2017*), we engineered an isogenic pair of STAG2-wildtype and -deficient HCT 116 colon carcinoma cells and homozygously inserted ARTi target sites (besides an AID::V5 tag; explained in the 'Materials and methods' section) into the 3'-UTR of endogenous *STAG1* (*Figure 2I*). Western blotting confirmed the knockout of *STAG2*, the insertion of the targeting cassette into the *STAG1* locus, and potent suppression of STAG1 following dox-induced ARTi-shRNAmir expression (*Figure 2J and K* and *Figure 2—figure supplement 3B*). Suppression of STAG1 in STAG2-deficient HCT 116 cells impaired their proliferation in vitro (*Figure 2—figure supplement 3C*) and the progression of xenografted tumors in vivo (*Figure 2L*), while dox-induced expression of the ARTi-shRNAmir had no antiproliferative effects in the presence of STAG2.

## Discussion

Together, these studies establish ARTi as a versatile and precise LOF method for target validation in oncology and beyond. Instead of designing gene-specific LOF reagents that remain prone to off-target effects, ARTi involves a simple, highly standardized experimental procedure that provides full control over on- and off-target effects and can be applied to any coding and non-coding gene. In cells that are pre-engineered to express the dox-inducible ARTi-shRNAmir, candidate genes can be converted into ARTi target genes and subsequently evaluated head-to-head in a highly standardized manner, either through knocking in ARTi target sites into endogenous loci or through knockout rescue approaches. Placement of ARTi target sites in non-coding transcript regions leaves the endogenous target protein unaltered, which is a key advantage over chemical-genetic LOF methods relying on the introduction of degron tags (*Lai and Crews, 2017*; *Cromm and Crews, 2017*). Targeting endogenously engineered ARTi target sites through tetracycline (Tet)-inducible ARTi-shRNAmir expression enables inducible, titratable, and likely reversible (not tested in this study) LOF perturbations of candidate genes without altering their endogenous transcriptional regulation, which is not possible using

conventional Tet-expression systems. In principle, by engineering multiple target sites in the same cell, ARTi enables combinatorial LOF perturbations, for example, for modeling synergistic target interactions, without multiplying the risk for off-target effects. Although ARTi requires the design of gene-specific sgRNAs and genome engineering steps that can be non-trivial, the method enables researchers to reach well-interpretable, comparable, and unambiguous experimental results that can guide larger investments in targets of high medical interest. Beyond providing a scalable method for early target validation, we foresee that ARTi can be used to establish on-target benchmark phenotypes for guiding the development and optimization of inhibitory molecules.

## Materials and methods
### Design and cloning of ARTi shRNAs

To design pairs of artificial shRNAs and matching target sites that trigger effective and selective target suppression with minimal off-target effects, nucleotide composition of shRNAs that reach exceptionally high-performance scores in a well-established siRNA prediction tool (DSIR [Designer of Small Interfering RNA]; *Vert et al., 2006*) and contain no A or T in guide position 20 to eliminate shRNAmirs that produce RISC-loadable small RNAs from the passenger strand were analyzed. To establish the criteria for ARTi design, siRNA predictions for the human and mouse genome were retrieved from DSIR (*Vert et al., 2006*) using default parameters. Top-scoring predictions (DSIR score >105) harboring G or C in position 20 were analyzed for nucleotide biases, which were used to define basic design criteria at the 5′-end the 3′-half. Next, possible off-targets were minimized using siSPOTR (*Boudreau et al., 2013*), an siRNA-based prediction tool that assesses off-target potential of different siRNA seed sequences (guide positions 2–8) in the human and mouse genome. Seed sequences with the lowest predicted off-target activity (top 1%) showed biases for C and/or G in guide positions 2–6, but 23% contained a T at the 5′ end of the seed sequence that is required in our design (*Figure 1B*). Among these, we observed particularly strong biases for CG in the following two positions (*Figure 1B*), which were found in 73% of all seed sequences harboring a 5′ T. Overall, 17% of all top-scoring seed sequences harbored TCG at their 5′ end, making it the second most common triplet (after CGC, which was found in 20%). Based on these analyses, we fixed the first four nucleotides of the guide to TTCG. For the following positions, we reasoned that introducing additional GC biases would destroy 5′–3′ asymmetry of small RNA duplexes that is critically required for efficient RISC loading. To maintain sufficient asymmetry, we therefore decided to bias the next three positions toward A or T, which in most positions is in alignment with nucleotide biases associated with knockdown efficacy.

For the remaining sequence 3′ of the seed region, we adhered to nucleotide features associated with knockdown efficacy based on our DSIR analysis, which are remarkably prominent and cannot all be explained through established processing requirements. Altogether, this established a 22-nt matrix for the design of ARTi shRNAmirs (TTCGWWWWNNAHHWWCATCCGGN; W = A/T, H = A/T/C; N = A/T/G/C). In a last step, we further reduced possible off-target effects by eliminating all guides whose extended seed sequence (guide positions 2–14) had a perfect match in the human or mouse transcriptome and, finally, selected the following six top-scoring ARTi predictions for experimental validation:

ARTi.6588 – target site: TCCGGATGAAGTTTATATCGAA/shRNAmir (97mer):

> TGCTGTTGACAGTGAGCGCCCGGATGAAGTTTATATCGAATAGTGAAGCCACAGATGTATTC
> GATATAAACTTCATCCGGATGCCTACTGCCTCGGA

ARTi.6570 – target site: TCCGGATGATATTGTTATCGAA/shRNAmir (97mer):

> TGCTGTTGACAGTGAGCGCCCGGATGATATTGTTATCGAATAGTGAAGCCACAGATGTAT
> TCGATAACAATATCATCCGGATGCCTACTGCCTCGGA

ARTi.6634 – target site: TCCGGATGATGTTTTAATCGAA/shRNAmir (97mer):

> TGCTGTTGACAGTGAGCGCCCGGATGATGTTTTAATCGAATAGTGAAGCCACAGATGTAT
> TCGATTAAAACATCATCCGGATGCCTACTGCCTCGGA

ARTi.6786 – target site: TCCGGATGATATTGTATACGAA/shRNAmir (97mer):

> TGCTGTTGACAGTGAGCGCCCGGATGATATTGTATACGAATAGTGAAGCCACAGATGTAT
> TCGTATACAATATCATCCGGATGCCTACTGCCTCGGA

ARTi.6834 – target site: TCCGGATGATATTGCATACGAA/shRNAmir (97mer):

TGCTGTTGACAGTGAGCGCCCGGATGATATTGCATACGAATAGTGAAGCCACAGATGTAT
TCGTATGCAATATCATCCGGATGCCTACTGCCTCGGA

ARTi.6516 – target site: TCCGGATGAAGTTTAATTCGAA/shRNAmir (97mer):

TGCTGTTGACAGTGAGCGCCCGGATGAAGTTTAATTCGAATAGTGAAGCCACAGATGTAT
TCGAATTAAACTTCATCCGGATGCCTACTGCCTCGGA

The following ARTi target sequences were used for experimental validation studies:
Insertion into the coding sequence before the STOP codon:

ATCCGGATGATATTGTATACGAATCCGGATGATATTGTTATCGAA (with the first 'A' being inserted to retain the reading frame)

Insertion after the STOP codon:

TCCGGATGATATTGTATACGAATCCGGATGATGTTTTAATCGAATCCGGATGATATTGTTATCG
AA

shRNAs were ordered as single-stranded DNA Ultramer oligonucleotides (Integrated DNA Technologies), amplified by PCR and cloned into different retroviral or lentiviral miRE/miRF shRNAmir expression vectors (LT3GFPIR *Fellmann et al., 2013*) using EcoR/XhoI restriction digest or Gibson assembly.

## Cell culture

Human HCT 116 cells (ATCC: CCL-247) were cultured in RPMI 1640 medium (Thermo Fisher) and PC-9 cells in McCoy's 5A medium (Thermo Fisher), supplemented with 10% FBS and 1x GlutaMAX (Thermo Fisher). RKO (ATCC: CRL-2577) and MOLM-13 cells (DSMZ: ACC 584) were cultured in RPMI 1640, supplemented with 10% FBS (Sigma-Aldrich), 4 mM L-glutamine (Thermo Fisher), 1 mM sodium pyruvate (Sigma-Aldrich), and penicillin/streptomycin (100 U ml$^{-1}$/100 µg ml$^{-1}$, Sigma-Aldrich). HT-1080 (ATCC: CCL-121) and LentiX lentiviral packaging cells (Clontech, Cat# 632180) were cultivated in DMEM (Thermo Fisher) with 10% FBS, 4 mM L-glutamine, 1 mM sodium pyruvate, and penicillin/streptomycin (100 U ml$^{-1}$/100 µg ml$^{-1}$). MV4-11 cells (ATCC: CRL-9591) were cultured in IMDM with 10% FBS, 4 mM L-glutamine, 1 mM sodium pyruvate, and penicillin/streptomycin (100 U ml$^{-1}$/100 µg ml$^{-1}$).

Murine MLL-AF9$^{OE}$, Nras$^{G12D}$ AML cells (RN2; *Zuber et al., 2011*) were cultured in RPMI 1640 medium supplemented with 10% FBS, 20 mM L-glutamine, 10 mM sodium pyruvate, 10 mM HEPES (pH 7.3), penicillin/streptomycin (100 U ml$^{-1}$/100 µg ml$^{-1}$), and 50 µM β-ME. *Kras$^{G12D}$*, *Trp53$^{-/-}$*, *MYC$^{OE}$* PDAC cells (EPP2), SV40 large T antigen immortalized mouse embryonic fibroblasts (RRT-MEF; *Fellmann et al., 2011*), and NIH/3T3 (ATCC: CRL-1658) were cultured in DMEM supplemented with 10% FBS, 20 mM glutamine, 10 mM sodium pyruvate, and penicillin/streptomycin (100 U ml$^{-1}$/100 µg ml$^{-1}$). MIA PaCa-2 (ATCC: CRL-1420) and GP2d (Ecacc: 95090714) cells were cultured in DMEM supplemented with 10% FBS. Ba/F3 (DSMZ: ACC300) cells were cultured in RPMI 1640 medium supplemented with 10% FBS, 10 ng ml$^{-1}$ IL-3 (R&D Systems), and Ls513 cells were cultured in RPMI 1640 medium supplemented with 10% FBS. All cell lines were maintained at 37°C with 5% CO$_2$, routinely tested for mycoplasma contamination, and authenticated by short tandem repeat analysis.

## Reporter assay

SFFV-GFP-P2A-Puro-ARTi-target sensor was cloned into pRSF91 retroviral plasmid (*Galla et al., 2011*) using Gibson assembly. RRT-MEFs were transduced with retroviruses expressing a GFP-reporter harboring the target sites for validated shRNAs and one ARTi-shRNA in its 3'-UTR. For each reporter cell line, single cells were FACS-sorted into 96-well plates using a FACSAria III cell sorter (BD Biosciences) to obtain single-cell-derived clones. These clones were transduced with retrovirus constructs in pSin-TRE3G-mCherry-miRE-PGK-Neo (TCmPNe) backbone expressing either the respective dox-inducible ARTi shRNA or validated shRNAs and mCherry fluorescence marker. shRNA expression was induced with dox and GFP levels were quantified via flow cytometry 2 d post induction. Knockdown efficiency was calculated as 1 minus the ratio of mean GFP signal in mCherry$^+$ (shRNA$^+$) cells over mCherry$^-$ cells and normalized to Renilla luciferase specific neutral control shRNA (Ren.713).

## Competitive proliferation assay

To investigate the effect of ARTi constructs in the absence of the endogenous target gene, competitive proliferation assays were performed in several human and murine cell lines. Human HT-1080, RKO, MOLM-13, and MV4-11 cell lines were lentivirally transduced with shRNAmir expression constructs cloned into pRRL-SFFV-GFP-miRF-PGK-Neo (SGFN) backbone at 20–60% efficiency. Initial infection efficiency was determined at day 4 post transfection (day 0) by measuring GFP expression as a readout using iQue Screener Plus flow cytometer (IntelliCyt). Percentage of shRNA+ cells (GFP positive) was monitored by flow cytometry in regular intervals, and results were normalized to day 0.

Human GP2d, Ls513, and MIA PaCa-2 cell lines were lentivirally transduced with shRNAmir constructs cloned into pRRL-TRE3G-GFP-miRE-PGK-Puro-IRES-rtTA3 backbone (LT3GEPIR, Addgene plasmid #111177). 500 cells were seeded in duplicates in 96-well plates and treated with 1 µg ml⁻¹ dox for 9–10 d and analyzed with Incucyte (Sartorius). Untreated cells served as reference.

Murine NIH-3T3, EPP2, and RN2 cells were retrovirally transduced with shRNAmir constructs cloned into pMSCV-miR-E-PGK-Neo-IRES-mCherry backbone (LENC; Addgene plasmid #111163), and initial infection levels were determined by flow cytometry based on mCherry expression 4 d post transduction (day 0).

## Crystal violet staining

To visualize ARTi-shRNAmir's effect, crystal violet staining assays were performed. 25,000 HCT 116 cells and 15,000 PC-9 cells per well were seeded in a 6-well plate containing 2 ml tetracycline-free growth medium and 1 µg ml⁻¹ dox. Medium was exchanged every 2–3 d. After 9 d, wells were washed with ice-cold PBS and subsequently stained with 1 ml of 2.3% crystal violet solution for 10 min. Subsequently, wells were washed with ultrapure water and dried overnight. Images were obtained with a scanner.

## Transcriptional profiling

For the unbiased identification of ARTi shRNA off-targets, RKO and HT-1080 shRNA+ cells from the competition proliferative assay experiment were selected with Geneticin/G418 Sulfate (Gibco) for 7 d and checked for GFP expression. On day 7, one arm of the empty vector control group was treated with IC₅₀ concentration of trametinib (MedChem Express: HY-10999) for 24 hr based on the data in the Genomics of Drug Sensitivity in Cancer database (https://www.cancerrxgene.org; *Yang et al., 2013*). Subsequently, cells were trypsinized, washed with ice-cold PBS, pelleted, and snap frozen. Total RNA was isolated using in-house magnetic beads kit and King Fisher Duo Prime Purification System (Thermo Fisher). NGS libraries were prepared with QuantSeq 3' mRNA-Seq Library Prep Kit (FWD) HT for Illumina (Lexogen) and UMI Second Strand Synthesis Module for QuantSeq FWD (Lexogen). Samples were sequenced on Illumina NovaSeq platform with 100 bp single-read protocol.

Engineered MIA PaCa-2 and PC-9 cells were cultured in the presence of 1 µg ml⁻¹ dox to induce expression of the ARTi shRNA. dox-containing media were replenished twice weekly and on days 4 and 8 after the initial treatment. $2 \times 10^6$ dox-treated and untreated control cells were harvested, washed with PBS, lysed, treated with DNAse I (QIAGEN), and total RNA was extracted using the RNeasy Mini Kit (QIAGEN). NGS libraries were prepared as above. Samples were sequenced on an Illumina NextSeq 2000 platform with a 75 bp protocol.

## Bioinformatic analyses of 3′ mRNA-seq

For 3′ mRNA-seq reads derived from the human cell lines HT-1080 and RKO, the 6-nt-long 5′ UMIs were attached to each read name with umi-tools (v1.0.0; *Smith et al., 2017*). Subsequently, the UMIs plus the next four nucleotides (UMI spacer), as well as 3′ adapters (stringency of 3) and bases with low quality (threshold of 25) were trimmed away using cutadapt (v1.18; *Martin, 2011*) and its wrapper tool trimgalore (v0.6.2). Read quality control was performed with FastQC (v0.11.8). The remaining reads were sample-wise aligned to the human (GRCh38.p13; GCA_000001405.28) reference genome. Mapping and subsequent filtering of 3′-UTR mapped reads was performed with slamdunk (v0.4.3; *Neumann et al., 2019*) in QuantSeq mode (slamdunk map -5 12 -n 100 -q). 3′ UTR regions were assembled based on the description in *Muhar et al., 2018*. Aligned and filtered reads were deduplicated with umi-tools (v1.0.0) (*Housden et al., 2017*), based on the mapping coordinate and the UMI attached to the read name, prior to quantifying read abundances within 3′-UTR regions using feature

Counts (v2.0.1; *Liao et al., 2014*). Differential expression analysis (DEA) was performed with DESeq2 (v1.30.1; *Love et al., 2014*) for each ARTi shRNA to empty vector control. Here, the number of up- and downregulated genes were calculated by filtering the DEA results for genes with a log2 fold-change ≥2 (up) or ≤2 (down) and a -log10 p-value ≥5. Principal component analysis was performed on the 1000 most variable expressed genes with the prcomp function from the stats (v4.2.0) R package.

For transcriptional profiling of the human cell lines MIA PaCa-2 and PC-9, the genome reference file and annotations were constructed based on the GRCh38 assembly and the Ensembl 86 version, respectively. The sequences of the shRNA construct (ARTi.6570 [RN_v_76, RN_v_118]) as well as the EGFR$^{del19}$::V5::dsRed::ARTi (RN_v_108) and the dsRed::Linker::KRAS$^{G12R}$-ARTi (RN_v_287) were also included. Mapping of the sequencing reads derived from the human cell lines was performed with STAR (v2.5.2b; *Dobin et al., 2013*) aligner allowing for soft clipping of adapter sequences. Quantification of read counts to transcript annotations was implemented using RSEM (v1.3.0; *Li and Dewey, 2011*) and featureCounts (v1.5.1; *Liao et al., 2014*). Normalization of read counts and differential analysis was implemented with the limma (*Ritchie et al., 2015*) and voom (*Law et al., 2014*) R packages.

## Genome engineering of EGFR in PC-9 and Ba/F3

Genome engineering of PC-9 cells was done as previously described (*Wilding et al., 2022*). In brief: PC-9_RIEN cells were transduced with an ecotropic pMSCV-EGFRdel19_V5_dsRed_ARTi-PGK-Blasticidin retrovirus cloned at GenScript and produced in Platinum E cells (Cell Biolabs) in the presence of 8 µg ml$^{-1}$ Polybrene (Merck Millipore). After 24 hr, stable transgenic cell pools were selected using 10 µg ml$^{-1}$ Blasticidin (Sigma-Aldrich). Subsequently, cells were diluted to obtain single cell clones. After 14 d of culture, single cell-derived colonies were transferred to 6-well plates and analyzed by western blot. Identified homozygous PC-9_ EGFR$^{del19}$-ARTi clones were further engineered by cutting endogenous EGFR with a CRISPR all-in-one vector pX458_Exon20_gRNA TAGTCCAGGAGGCAGC CGAA (GenScript) using X-tremeGENE 9 DNA transfection reagent (Roche) according to the protocol supplied by the vendor. 48 hr after transfection, GFP-positive cells were sorted by FACS (SONY cell sorter S800Z) and diluted to obtain single-cell clones. Positive clones, which contained only the exogenous EGFR$^{del19}$-ARTi, but not the endogenous EGFR, were identified by western blot. Next, the selected EGFR clone was transduced with a pantropic LT3GEPIR_Puro_ARTi-shRNA TTCGATAA CAATATCATCCGGA retrovirus cloned at GenScript, China, and produced via the Lenti-X Single Shot system (Clontech). 72 hr later, stable transgenic cell pools were selected using 0.5 µg ml$^{-1}$ Puromycin (Sigma-Aldrich). Following the selection, cells were diluted to obtain single cell clones. After 14 d of culture, single cell-derived colonies were transferred to 6-well plates and induced via 1 µg ml$^{-1}$ dox. Positive clones were characterized by a strong GFP-induction that was identified by flow cytometry.

Ba/F3 cells were transduced with an ecotropic pMSCV-EGFRdel19_V5_dsRed_ARTi-PGK-Blasticidin retrovirus cloned at GenScript and produced in Platinum E cells in the presence of 4 µg ml$^{-1}$ Polybrene. After 72 hr, stable transgenic cells were selected by using 50 µg ml$^{-1}$ Blasticidin, without adding IL-3.

## Genome engineering of KRAS in MIA PaCa-2

MIA PaCa-2 ARTi-shRNAmir-expressing cells were transduced with an ecotropic pMSCV-dsRed::KRAS$^{G12R}$-ARTi-PGK-Blasticidin retrovirus cloned at GenScript, China, and produced in Platinum E cells in the presence of 8 µg ml$^{-1}$ Polybrene. After 24 hr, stable transgenic cell pools were selected using 10 µg ml$^{-1}$ Blasticidin. Subsequently, endogenous KRAS was knocked out by transient transfection of three gRNAs targeting exon 2, and the region containing the G12C variant (present in MIA PaCa-2 cells) was used in a co-transfection (gRNA#3: *GAATATAAACTTGTGGTAGT*; gRNA#6: *CTTGTGGTAGTTGGACTTG*; gRNA#7: *GTAGTTGGAGCTTGTGGCGT*). Knockout clones were identified by the absence of the endogenous KRAS protein using western blot.

## Genome engineering of STAG1/STAG2 in HCT 116

HCT 116 cell line was engineered by cutting STAG1 with gRNAs targeting the region close to the STOP codon. Guide RNAs *TTCTTCAGACTTCAGAACAT* or *CTGAAGAAAATTTACAAATC* were cloned into the pSpCas9(BB)–2A-GFP plasmid (pX458; Addgene plasmid 48138) and used in a co-transfection. Simultaneously, a STAG1_AID_V5_P2A_Blasti_STOP_ARTi repair template with 800 bp of left and right homologous arms of the STAG1 genomic locus (in pUC57-Simple backbone) was transfected into HCT 116 cells using Lipofectamine 3000 transfection reagent (Thermo Fisher) according to the

manufacturer's instructions. A stable transgenic cell pool was selected 48 hr after transfection using 5 µg ml$^{-1}$ Blasticidin and diluted to obtain single-cell clones. Positive clones were identified by western blot.

Identified homozygous HCT 116_STAG1_ARTi clones were further engineered by disrupting STAG2 gene with a CRISPR all-in-one vector Hs0000077505_U6gRNA-Cas9-2A-GFP and Hs0000077502_U6gRNA-Cas9-2A-GFP, respectively. Cells were transfected using X-tremeGENE 9 DNA transfection reagent according to the manufacturer's instructions, sorted 48 hr post transfection for GFP-positive cells, and diluted to obtain single-cell clones. Positive clones were identified by western blot.

Next, the selected HCT 116 clone was transduced with a pantropic LT3GEPIR_Puro_ARTi-shRNA *TTCGATAACAATATCATCCGGA*retrovirus cloned at GenScript, China, and produced via the Lenti-X Single Shot system (Clontech). 72 hr later, stable transgenic cell pools were selected using 2 µg ml$^{-1}$ Puromycin (Sigma, P9620). Following the selection, cells were diluted to obtain single-cell clones. After 14 d of culture, single cell-derived colonies were transferred to 6-well plates and induced via 1 µg ml$^{-1}$ dox. Positive clones were characterized by a strong GFP-induction that was identified by flow cytometry.

## Western blot

The following primary antibodies were used for immunoblot analyses: EGFR (Cell Signaling, #4267); STAG1 (GeneTex, GTX129912); STAG2 (Bethyl, A300-159A); b-actin (Sigma, A5441); KRAS (LSbio, LS-C17566); and V5 (Sigma, V8012). PC-9, MIA PaCa-2, and HCT 116 cell pellets harboring EGFR, KRAS, and STAG1/STAG constructs respectively were lysed in Triton X-100 lysis buffer, sonicated, and stored at –80°C. For protein detection, the pellets were thawed on ice, followed by 15 min centrifugation at 13,000 rpm and 4°C. Furthermore, cell lysates were loaded onto a pre-casted SDS–polyacrylamide gel (4–12%) and proteins were transferred onto a nitrocellulose or PVDF membrane. Membranes were probed with the respective primary antibodies overnight. The next day, secondary antibodies conjugated with fluorescent dye were added and the proteins were detected by the Odyssey detection system.

## Compound treatment

To investigate sensitivity to EGFR-targeting compounds, cell viability was determined using the Cell Titer Glo assay (Promega). For this purpose, 10 mM stock solutions in DMSO of afatinib (*Li et al., 2008*), osimertinib (*Liu et al., 2018*; *Soria et al., 2018*; *Liu et al., 2018*), and poziotinib (*Kim et al., 2019*; *Robichaux et al., 2019*) were used. 5000 cells per well were seeded in 150 µl of the medium in technical triplicates in 96-well plates and incubated at 37°C and 5% $CO_2$ for 5 hr, followed by the compound addition. Cells were treated with seven different concentrations of inhibitors in a serial eightfold dilutions starting with the highest concentration of 3 µM. For comparability, DMSO normalization to the highest added volume was performed. Subsequently, cells were cultivated for 96 hr at 37°C and 5% $CO_2$. 50 µl of Cell Titer Glo reagent was added to each well, incubated for 10 min in the dark, and luminescence was measured using the multilabel Plate Reader VICTOR X4. The measurement time was set to 0.2 s. Luminescence values relative to DMSO-treated cells were plotted in GraphPad Prism and fitted using nonlinear regression with a variable slope to calculate IC50 values at 50% inhibition. MIA PaCa-2 KRAS G12C inhibitor (AMG-510) (*Fakih et al., 2020*; *Lanman et al., 2020*; *Fakih et al., 2019*) treatments were performed as described for EGFR with the following modifications: 2000 cells per well and 0.5 nM to 3 µM concentration range of AMG-510.

## In vivo experiments

The PC-9 EGFR k.o., EGFR[del19]::ARTi study was performed at the AAALAC-accredited animal facility of CrownBio Leicestershire, UK. Female NSG (NOD.Cg-PrkdcscidII2rgtm1wjl/SzJ) Crl mice were obtained from Charles River. The age of animals at study initiation was 7–8 wk and had an acclimatization period of ≥14 d. Mice were group-housed in IVCs. The study complies with the UK Animals Scientific Procedures Act 1986 (ASPA) in line with Directive 2010/63/EU of the European Parliament and the Council of September 22, 2010, on the protection of animals used for scientific purposes.

All other in vivo experiments were performed at the AAALAC-accredited animal facility of Boehringer Ingelheim RCV GmbH & CoKG. Female BomTac:NMRI-*Foxn1[nu]* mice were obtained from Taconic Denmark at 6–8 wk of age. After the arrival, mice were allowed to adjust to the housing

conditions at least for 5 d before the start of the experiment. Mice were housed in pathogen-free and controlled environmental conditions (open-cage housing), and handled according to the institutional, governmental, and European Union guidelines (Austrian Animal Protection Laws, GV-SOLAS and FELASA guidelines).

Studies were approved by the internal ethics committee (called 'ethics committee') of Boehringer Ingelheim RCV GmbH & Co KG in the Department of Cancer Pharmacology and Disease Positioning. Furthermore, all protocols were approved by the Austrian governmental committee (MA 60 Veterinary office; approval numbers GZ: 903122/2017/21 and GZ: 416181-2020-29).

To establish subcutaneous tumors, mice were injected with $2 \times 10^6$ HCT 116 in PBS, $5 \times 10^6$ MIA PaCa-2 in 1:2 Matrigel:PBS with 5% FBS or with $1 \times 10^7$ PC-9 cells in PBS with 5% FBS.

Tumor diameters were measured with a caliper three times a week. The volume of each tumor (in mm³) was calculated according to the formula 'tumor volume = length * diameter² * π/6.' Mice were randomized into the treatment groups when tumor size reached between ~130 and 190 mm³. Group sizes were calculated for each tumor model based on tumor growth during model establishment experiments. A power analysis was performed using a sample size calculator (https://www.stat.ubc.ca/~rollin/stats/ssize/n2.html). For all models used in the studies, 10 mice per group were used. 2 mg ml⁻¹ doxycycline hyclate (Sigma) and 5 mg ml⁻¹ sucrose were added to the drinking water of the treatment groups, the control group received water with 5 mg ml⁻¹ sucrose only. Osimertinib (Tagrisso, AstraZeneca, 40 mg tablet) was dosed per os daily at a dose of 25 mg kg⁻¹ in Natrosol, and control mice were dosed per os daily with Natrosol. To monitor side effects of treatment, mice were inspected daily for abnormalities and body weight was determined three times per week. Animals were sacrificed when the tumors reached a size of 1500 mm³. Food and water were provided ad libitum. In vivo experiments were not repeated.

## Immunohistochemistry

Xenograft samples were fixed in 4% formaldehyde for 24 hr and embedded in paraffin. 2-μm-thick sections were cut using a microtome. STAG1 was stained with Polink 2 Plus HRP rat NM detection system (GBI Labs #D46-6) according to the manufacturer's instructions using a recombinant rat monoclonal STAG1 antibody (Abcam ab241544, Lot:GR3334172-1; 1:200) after cooking 10 min at 110°C in antigen unmasking solution (Vector #H3301). After staining, the slides were digitalized (scanner: Leica Aperio AT2). All slides were reviewed and evaluated for quality by a board-certified MD pathologist. Imaging analysis was performed using the digital pathology platform HALO (Indica Labs). A tissue-classifying algorithm was trained to selectively recognize viable tumor tissue against stroma, necrosis, and skin. The tissue classification output for each scan was reviewed and manually edited as necessary. A cell detection and scoring algorithm was trained to measure DAB optical density (OD) in the nuclei of tumor cells. A positivity threshold for DAB OD was determined by normalization with respect to the DAB OD as calculated from bona fide negative tissue (e.g., murine stroma as background). The object data for background-normalized nuclear ODs for each tumor cell were exported from three control and three doxycycline-treated cases (207945 and 33331 pooled objects, respectively). The cumulative distribution of the background-normalized DAB OD for each tumor cell nucleus was then plotted for control and doxycycline-treated cases, separately (*Figure 2K*).

## Cell lines

All newly generated ARTi cell lines are available upon request. The following parental cell lines were used:

| Cell line name | Species | Supplier name | Catalog number | Clone number |
| --- | --- | --- | --- | --- |
| PC-9 | Human | ECACC | 90071810 | Lot# 14A030 |
| MIA PaCa-2 | Human | ATCC | CRL-1420 | Lot# 1350798 |
| HT 1080 | Human | ATCC | CCL-121 | Lot# 63835574 |
| HCT 116 | Human | ATCC | CCL-247 | Lot# 6028631 |
| RKO | Human | ATCC | CRL-2577 | Lot# 59354083 |

*Continued on next page*

*Continued*

| Cell line name | Species | Supplier name | Catalog number | Clone number |
|---|---|---|---|---|
| MV4-11 | Human | ATCC | CRL-9591 (old number) | Lot# 58352230 |
| MOLM-13 | Human | DSMZ | ACC 554 | Lot# 11 |
| NIH-3T3 | Human | ATCC | CRL-1658 | Lot# 63292276 |
| EPP2 | | KrasG12D, Trp53-/-, MYCOE PDAC cells (EPP2) | NA | NA |
| RN2 | Murine | MLL-AF9OE, NrasG12D AML cells (RN2; *Zuber et al., 2011*) | NA | NA |
| Gp2d | Human | ECACC | 95090714 | Lot# 11E013 |
| Ls513 | Human | ATCC | CRL-2134 | Lot# 57761342 |
| Ba/F3 | Murine | DSMZ | ACC300 | |
| 293T-Lenti-X | Human | Clontech | 632180 | Lot# 1404558A |

## Acknowledgements

The authors thank all colleagues in the labs of JZ and RAN for critical discussions and technical support. JZ was supported by the Swiss National Science Foundation (Early Postdoc Mobility Fellowship P2BSP3_188110) and European Union's Horizon 2020 research and innovation program under the Marie Skłodowska-Curie grant agreement no. 847548 (VIP2) and no. 101032582 (AML-SynergyX). Research at the IMP was supported by Boehringer Ingelheim and the Austrian Research Promotion Agency (headquarter grant FFG-852936). MS is a member of the Boehringer Ingelheim Discovery Research global post-doc program.

## Additional information

### Competing interests

Thomas Hoffmann: Full-time employee of Advantage Therapeutics GmbH. Alexandra Hörmann, Maja Corcokovic, Jasko Salkanovic, Fiona Spreitzer, Anna Köferle, Katrin Gitschtaler, Alexandra Popa, Sarah Oberndorfer, Nicole Budano, Jan G Ruppert, Paolo Chetta, Melanie Wurm, Ralph A Neumüller: Full time employee Boehringer Ingelheim. The other authors declare that no competing interests exist.

### Funding

| Funder | Grant reference number | Author |
|---|---|---|
| Swiss National Science Foundation | P2BSP3_188110 | Jakub Zmajkovic |
| Austrian Research Promotion Agency | FFG-852936 | Johannes Zuber |
| Marie Skłodowska-Curie Actions | 847548 (VIP2) | Jakub Zmajkovic |
| Marie Skłodowska-Curie Actions | 101032582 (AML-SynergyX) | Jakub Zmajkovic |

The funders had no role in study design, data collection and interpretation, or the decision to submit the work for publication.

### Author contributions

Thomas Hoffmann, Alexandra Hörmann, Maja Corcokovic, Jakub Zmajkovic, Matthias Hinterndorfer, Investigation, Writing – original draft; Jasko Salkanovic, Fiona Spreitzer, Anna Köferle, Katrin Gitschtaler, Sarah Oberndorfer, Florian Andersch, Markus Schaefer, Michaela Fellner, Nicole Budano,

Jan G Ruppert, Paolo Chetta, Melanie Wurm, Investigation; Alexandra Popa, Formal analysis; Johannes Zuber, Conceptualization, Supervision, Investigation, Visualization, Writing – original draft, Project administration, Writing – review and editing; Ralph A Neumüller, Conceptualization, Formal analysis, Supervision, Visualization, Writing – original draft, Project administration, Writing – review and editing

## Author ORCIDs
Thomas Hoffmann  http://orcid.org/0000-0002-1589-0013
Jakub Zmajkovic  http://orcid.org/0000-0001-7061-6538
Matthias Hinterndorfer  http://orcid.org/0000-0003-2435-4690
Anna Köferle  http://orcid.org/0000-0003-1601-7774
Johannes Zuber  http://orcid.org/0000-0001-8810-6835
Ralph A Neumüller  http://orcid.org/0000-0002-1514-6278

## Ethics
Studies were approved by the internal ethics committee of Boehringer Ingelheim RCV GmbH & Co KG in the Department of Cancer Pharmacology and Disease Positioning. Furthermore, all protocols were approved by the Austrian governmental committee (MA 60 Veterinary office; approval numbers GZ: 903122/2017/21 and GZ: 416181-2020-29).

Reviewer #1 (Public Review): https://doi.org/10.7554/eLife.84792.3.sa1
Reviewer #2 (Public Review): https://doi.org/10.7554/eLife.84792.3.sa2
Author Response: https://doi.org/10.7554/eLife.84792.3.sa3

# Additional files

## Supplementary files
• MDAR checklist

## Data availability
The workflow for the RNA-seq bioinformatics analyses is described in the Materials and Methods section. RNA sequencing data were uploaded to GEO with the accession numbers: GSE218404 and GSE218617. The data will be made publicly available upon acceptance of this manuscript. All ARTi cell lines described in this study are available upon request.

The following datasets were generated:

| Author(s) | Year | Dataset title | Dataset URL | Database and Identifier |
|---|---|---|---|---|
| Zuber J | 2023 | Precision RNAi using synthetic shRNAmir target sites | http://www.ncbi.nlm.nih.gov/geo/query/acc.cgi?acc=GSE218404 | NCBI Gene Expression Omnibus, GSE218404 |
| Zuber J | 2023 | Precision RNAi using synthetic shRNAmir target sites | http://www.ncbi.nlm.nih.gov/geo/query/acc.cgi?acc=GSE218617 | NCBI Gene Expression Omnibus, GSE218617 |

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
