## [Editor Report · eLife assessment]

This manuscript describes a **valuable** method to study the mechanism of action of essential genes and novel putative drug targets. Evidence for the effectiveness of the system, which is based on engineering pre-validated targets for RNA-mediated knockdown into genes of interest, is **compelling**, and the method should find use as an orthogonal method for generating gene-specific knockdowns.

---

## [Referee Report · Reviewer #1 (Public Review)]

In this manuscript the authors proposed a novel system by which they can suppress the expression of any gene of interest precisely and efficiently with a pre-validated, highly specific and efficient synthetic short-hairpin RNA. The idea of identifying potent artificial RNAi (ARTi) triggers is intriguing, and the authors successfully identify six ARTi with robust knockdown efficiency and limited to no off-target effects. As a proof-of-concept, the authors examined three oncology targets for validation, including EGFRdel19 (which already has a clinically approved drug for validation), KRASG12R (for which there are no in vivo compatible inhibitors yet) and STAG1 (which has a synthetic lethal interaction with recurrent loss-of-function mutations of STAG2). The authors demonstrated significant suppression of colony formation and in vivo tumor growth for all three oncology targets.

This novel system could serve as a powerful tool for loss-of-function experiments that are often used to validate a drug target. Not only this tool can be applied in exogenous systems (like EGFRdel19 and KRASG12R in this paper), the authors successfully demonstrated that ARTi can also be used in endogenous systems by CRISPR knocking in the ARTi target sites to the 3'UTR of the gene of interest (like STAG2 in this paper).

ARTi enables specific, efficient, and inducible suppression of these genes of interest, and can potentially improve therapeutic target validations. However, the system cannot be easily generalized as there are some limitations in this system:

• The authors claim in the introduction that CRISPR/Cas9-based methods are associated with off-target effects, however, the author's system requires the use CRISPR/Cas9 to knock out a given endogenous genes or to knock-in ARTi target sites to the 3' UTR of the gene of interest. Though the authors used a transient CRISPR/Cas9 system to minimize the potential off-target effects, the methods does not, as the authors acknowledge, eliminate the possibility of off-target effects.

• Instead of generating gene-specific loss-of-function triggers for every new candidate gene, the authors identified a universal and potent ARTi to ensure standardized and controllable knockdown efficiency. It seems this would save time and effort in validating each lost-of-function siRNAs/sgRNAs for each gene. However, users will still have to design and validate the best sgRNA to knock out endogenous genes or to knock in ARTi target sites by CRISPR/Cas9. The latter is by no-means trivial. Users will need to design and clone an expression construct for their cDNA replacement construct of interest, which will still be challenging for big proteins. This is not, as the authors point out, a replacement for other LOF methods, and there are other ways to achieve gene-specific regulation via, for example, degrons. However it is an effective orthogonal approach that many users may find compelling for their applications.

---

## [Referee Report · Reviewer #2 (Public Review)]

In this manuscript, Hoffmann et al. introduce a novel and innovative method to validate and study the mechanism of action of essential genes and novel putative drug targets. In the wake of many functional genomics approaches geared towards identifying novel drug targets or synthetic lethal interactions, there is a dire need for methods that allow scientists to ablate a gene of interest and study its immediate effect in culture or in xenograft models. In general, these genes are lethal, rendering conventional genetic tools such as CRISPR or RNAi inept.

The ARTi system is based on expression of a transgene with an artificial RNAi target site in the 3'-UTR as well as a TET-inducible miR-E-based shRNAi. Using this system, the authors convincingly show that they can target strong oncogenes such as EGFRdel19 or KRasG12 as well as synthetic lethal interactions (STAG1/2) in various human cancer cell lines in vivo and in vitro.

The system is very innovative, likely easy to be established and used by the scientific community and thus very meaningful.

---

## [Author Response]

The following is the authors' response to the original reviews.

**Reviewer #1 (Public Review):**
[…] This novel system could serve as a powerful tool for loss-of-function experiments that are often used to validate a drug target. Not only this tool can be applied in exogenous systems (like EGFRdel19 and KRASG12R in this paper), the authors successfully demonstrated that ARTi can also be used in endogenous systems by CRISPR knocking in the ARTi target sites to the 3'UTR of the gene of interest (like STAG2 in this paper).

We thank the referee for highlighting the novelty and potential of the ARTi system.

ARTi enables specific, efficient, and inducible suppression of these genes of interest, and can potentially improve therapeutic target validations. However, the system cannot be easily generalized as there are some limitations in this system:• The authors claimed in the introduction sections that CRISPR/Cas9-based methods are associated with off-target effects, however, the author's system requires the use CRISPR/Cas9 to knock out a given endogenous genes or to knock-in ARTi target sites to the 3' UTR of the gene of interest. Though the authors used a transient CRISPR/Cas9 system to minimize the potential off-target effects, the advantages of ARTi over CRISPR are likely less than claimed.

We thank the reviewer for raising these very valid concerns about potential off-target effects related to the CRISPR/Cas9-based gene knockout or engineering of endogenous ARTi target sites. In contrast to conventional RNAi- and CRISPR-based approaches, such off-target effects can be investigated prior to loss-of-function experiments through comparison between parental and engineered cells, which in the absence of CRISPR-induced off-target events are expected to be identical. Subsequent ARTi experiments provide full control over RNAi-induced off-target activities through comparison of target-site engineered and parental cells. However, we agree that undetected CRISPR/Cas9-induced off-target events cannot be ruled out in a definitive way, which we have pointed out in our revised manuscript.

• Instead of generating gene-specific loss-of-function triggers for every new candidate gene, the authors identified a universal and potent ARTi to ensure standardized and controllable knockdown efficiency. It seems this would save time and effort in validating each lost-of-function siRNAs/sgRNAs for each gene. However, users will still have to design and validate the best sgRNA to knock out endogenous genes or to knock in ARTi target sites by CRISPR/Cas9. The latter is by no-means trivial. Users will need to design and clone an expression construct for their cDNA replacement construct of interest, which will still be challenging for big proteins.

We fully agree that the required design of gene-specific sgRNAs and subsequent CRISPR-engineering steps are by no means trivial. However, we believe that decisive advantages of the method, in particular the robustness of LOF perturbations and additional means for controlling off-target activities, can make ARTi an investment that pays off. In our experience, much time can be lost in the search for effective LOF reagents, and even when these are found, questions about off-target activity remain. While ARTi overcomes many of these challenges by providing a standardized experimental workflow, we do not propose to replace all other LOF approaches by this method. Instead, we would position ARTi as a unique orthogonal approach for the stringent validation and in-depth characterization of candidate target genes, as we have highlighted in our revised discussion.

• The approach of knocking-out an endogenous gene followed by replacement of a regulatable gene can also be achieved using regulated degrons, and by tet-regulated promoters included in the gene replacement cassette. The authors should include a discussion of the merits of these approaches compared with ARTi.

We thank the reviewer for pointing out these alternative LOF methods. We had already included a brief discussion of chemical-genetic LOF methods based on degron tags. While we certainly share the current excitement about degron technologies, they inevitably require changes to the coding sequence of target proteins, which can alter their regulation and function in ways that are hard to control for. In our revised discussion, we have added a brief comparison to conventional tet-regulatable expression systems, which unlike ARTi require the use of ectopic tet-responsive promoters. Overall, we would position ARTi as an orthogonal tool that enables inducible and reversible LOF perturbations without changing the coding sequence and the endogenous transcriptional control of candidate target genes.

**Reviewer #2 (Public Review):**
[…] The ARTi system is based on expression of a transgene with an artificial RNAi target site in the 3'-UTR as well as a TET-inducible miR-E-based shRNAi. Using this system, the authors convincingly show that they can target strong oncogenes such as EGFRdel19 or KRasG12 as well as synthetic lethal interactions (STAG1/2) in various human cancer cell lines in vivo and in vitro.The system is very innovative, likely easy to be established and used by the scientific community and thus very meaningful.

We thank the reviewer for her/his enthusiasm about ARTi.

**Reviewer #1 (Recommendations For The Authors):**
• The authors claimed that ARTi enables specific, efficient, inducible, and reversible suppression of any gene of interest. However, there are no experiments supporting the reversible suppression of their gene of interest. Data are required to support this statement.

We thank the reviewer for pointing this out. The statement about the reversibility ARTi-mediated effects was based on a rich body of literature that has demonstrated the reversibility of Tet-shRNAmir-induced LOF perturbations and associated phenotypes. As ARTi employs the same Tet-shRNAmir expression vectors, we have no reason to believe that this feature would be lost. However, since we have not demonstrated this in our study, we have removed this statement in our revised manuscript.

• In Figure 1E, the authors did make the point by including trametinib treated samples as positive controls. However, the trametinib treated samples also made the transcriptome changes in the ARTi groups hard to read. I wonder what the PCA analysis will look like if the authors exclude the trametinib treated groups.

In Figure 1E, we used PCA as a common and easy-to-digest visualization tool to showcase the neutrality of ARTi shRNAmirs. Given the complete absence of significantly deregulated genes for all three ARTi shRNAmirs (Figure 1F), we believe that a PCA analysis of just these samples would merely represent experimental noise and not yield additional insights.

• This universal and potent ARTi should ensure standardized and controllable knockdown efficiency, however, the knockdown efficiency for KRASG12R is not as potent as that for EGFRdel19. The authors should discuss the differences.

We thank the reviewer for pointing this out. The exact level of knockdown on the protein level is hard to determine due to detection limits of the used method. The differences in the extent to mRNA knockdown could be attributable to cleavage efficiencies due to potential secondary structures in the respective mRNAs. We suspect that the KRASG12R transgene expresses at higher levels, compared to EGFRdel19. We might therefore still be looking at the same overall magnitude of knockdown. As we did not perform a detailed analysis of the respective knockdown levels, we do not feel comfortable in stating differences in knockdown levels and therefore do not think that addressing potential differences are justified.

• It is interesting to see that, unlike other cancer types, tumor burdens did not decrease after inducing knockdown of STAG1 in STAG2 knockout HCT116 lines in Figure 2L. Have the authors examined senescence markers in this set of mice?

We have not investigated these markers and thank the reviewer for this suggestion. More detailed analyses of the phenotype are planned.

• Have the authors carefully examined the transcriptome changes induced or if not across all targets at least in the case of ARTi knock into the 3'UTR of STAG1?

We thank the reviewer for this suggestion. This would indeed be interesting to conduct for STAG1/2, especially for genes with an integration of the ARTi into the 3’UTR. The reason why we did not perform this analysis with our cell lines is that we used a construct that also adds an AID tag to STAG1 (STAG1_AID_V5_P2A_Blasti_STOP_ARTi), as outlined in the methods section. After the engineering, STAG1 carries the ARTi sequence in the 3’UTR but is also fused to AID::V5. In addition a P2A::Blasticidin_resistance Protein is made from the same transcript. We chose to use this complex strategy with the aim of comparing AID mediated degradation with ARTi-mediated knockdown. Unfortunately, the AID-based approach did not work, and we were not able to observe a reduction in protein levels. We however observed lower expression of STAG1 in the engineered versus the parental cells, likely caused by the tag, and consequently did not conduct gene expression analyses, as we would not be able to assess if transcriptome changes could be solely ascribed to the changes in the 3’UTR. The knockdown levels are hence only analyzed on the protein level.

**Reviewer #2 (Recommendations For The Authors):**
This is a fantastic paper, easy to read and provides a very meaningful new and innovative approach for drug target validation. I think the manuscript could be further improved by adding a section to the discussion outlining other approaches that could be used to solve the same problem. For example, Bill Kaelin came up with a strategy of expressing shRNA- or sgRNA-resistant and rtTA- or tTA-regulated cDNAs of essential gene-of-interest followed by sh/sgRNA-mediated ablation of the endogenous gene (e.g.PMID: 28082722), which is conceptually quite similar to the ARTi approach. Similarly, people have used conditional degron tags such as AID tags, dTags, HALOTags, IHZF3 degrons or SMASh either knocked into the endogenous locus or as rescue transgene. Comparing and contrasting the pros and cons of these methods to the ARTi-based approach would be certainly beneficial to the readers.

We thank the referee for pointing out these alternative LOF methods. We certainly share the current excitement about various degron tags and are applying them in our own research. In our view, a major downside of these strategies is that they inevitably require changes to the coding sequence of target proteins, which can alter their regulation and function in ways that are hard to predict and control for. We had briefly mentioned this distinguishing feature in our discussion. The strategy proposed by Bill Kaelin, i.e. rescue of the the endogenous gene through Tet-regulated expression of sh/sgRNA-resistant cDNAs, indeed shares many features of the ARTi system, but requires expression of the candidate target from an ectopic promoter element. In contrast, ARTi enables similar perturbations of candidate genes without altering their endogenous transcriptional regulations – a feature that we have highlighted in our revised discussion.

All my other comments outlined below should be considered minor and are not essential.1, Suppl Fig.1 C: Please explain what the red star means. How can the knock-out be more than 100%. Please specify what the controls are. Why does shRNA660 exhibit no knockdown at all?

The red star indicates ARTi-shRNAmirs that were selected for further characterization. Depicted GFP knockdown levels are normalized to the performance of shRen.713, a well-characterized potent control shRNA targeting Renilla Luciferase. Values of more than 100% mean that the respective shRNA exceeded effects of shRNA.713. shRNA.660 served as a neutral control – its target site was not included in the reporter construct. We thank the reviewer for bringing up these points, which we have clarified in the legend.

2, x-axis label in Suppl Fig. 1D is missing

We thank the referee for spotting this and have added this information to the figure and its legend.

3, I would argue that ARTi6634 also has a slight effect in MV4-11 similar to its effect to RN2. Maybe add that to the text.

We thank the reviewer and have added this observation to our revised text.

4, Suppl. Figure Legend 1F - specify which cell line was used (HT-1080 presumably)

We apologize for this mistake and now have indicated the cell line in the legend.

5, Fig. 2A and E, it might be nice to add the dsRED fusion to the schematics so that the reader sees the difference between the endogenous and the endogenous. One could then also change the color to red instead of blue.

We thank the reviewer for this suggestion and adapted the figure accordingly.

6, Fig. 2B - In the third lane, there appears to be a residual band of the endogenous EGFR despite the fact that it should be KO. Is this a EGFR wt lysate with EGFR::dsRED::ARTi overexpression and as such a type in the legend or is this a non-complete KO? It might be beneficial to label the legend with EGFR::dsRED::ARTi instead of EGFR::ARTi have one arrow depicting EGFR and one additional arrow showing the EGFR::dsRED fusion (as in Fig. 1F).

We thank the reviewer for this insightful comment. We interpret the WB signal in lane three as potential cleavage/degradation products of the transgene as all signal disappears upon ARTi-mediated knockdown. Due to space reasons, we would prefer to keep the label as it is. The exact nature of the transgene is stated in the text and in the methods section.

7, Suppl Fig. 2d: It is interesting that there is such a huge upregulation of DUSP6 in cells that express EGFR::ARTi compared to parental? The figure legend states: expression levels of DUSP6 in parental and engineered PC-9 cells. I assume the first box (EGFR::ARTi -/ dox -) is the parental line? Is there really a 5x upregulation of DUSP6 upon overexpression of EGFR::ARTi compared to parental (despite the fact that the endogenous EGFR::ARTi is expressed to similar levels compared to the endogenous EGFR)? Please clarify a little better which of the cells are parental and which are EGFR KO and which are transduced with EGFR::ARTi. Might suffice to just explain in the supplmental figure legend that expression of the exogenous EGFR::ARTi in EGFR KO cells leads to increased expression of ERK targets such as DUSP6 and EPHA2 etc.

We thank the reviewer for this comment. We ascribe the increased expression of DUSP6 to the forced expression of the oncogenic variant of EGFR (EGFRdel19) while only a subset of *EGFR* genes in PC-9 cells is mutated and the rest is wild-type. Therefore, the net-output of EGFR signaling would be higher, even if the EGFR protein levels were exactly the same, as the *EGFR* gene is only present in the oncogenic form in the engineered cells but a mixture of mutant and wild-type proteins would make up the EGFR pool in the parental cells. The figure legend was changed accordingly, highlighting that *DUSP6* is a MAPK downstream gene.

8, Suppl Fig. 2e: Similar to my comment #7. Expression of endogenous EGFR is lost upon KO of EGFR, but cylcinD1 expression as well as expression of other ERK target genes increases upon loss of the endogenous EGFR gene with concomitant expression of EGFR::ARTi . It is nice to see that most of those genes are down-regulated upon DOX treatment. However, CyclinD1 is strongly up-regulated - any idea why? Might be nice to comment on this in the supplemental material to make it easy for the reader to interpret the data.

We agree with the reviewer that the direct MAPK target genes follow the expected pattern of strong downregulation. We have not studied the expression of CCND1 in detail and therefore cannot comment on the mechanistic basis of this observation.

9, Fig. 2F - might be nice to provide some densitometry data to quantify the effect of ARTi-mediated KRasG12R knock-down.

We thank the reviewer for this suggestion and apologize that this data is not available for this study. We will include densitometry data in upcoming studies involving ARTi. As the observed knockdown was almost complete and hence readily observable by eye, we did not measure the effects using densitometry. In addition, we would like to mention that the sensor assay contains a quantitative analysis of the knockdown levels.

10, Fig. 2I, it might be nice to add the V5 tag to the schematic and mention the V5 tag in the text: ... and homozygously inserted ARTi target sites into the 3'-UTR as well as a V5 tag to the endogenous STAG1 alleles (Fig. 2i)

We thank the reviewer for the suggestion and explained the exact makeup of the construct better in the main text. We would however like to keep the figure as simple as possible and put the focus on the endogenous engineering here.

11, Fig. 2J - might be nice to provide some densitometry data to quantify the effect of ARTi-mediated STAT1::V5 knock-down.

We thank the reviewer for this suggestion and apologize that this data is not available for this study. We will include densitometry data in upcoming studies involving ARTi. As the observed knockdown was almost complete and hence readily observable by eye, we did not measure the effects using densitometry. In addition, we would like to mention that the sensor assay contains a quantitative analysis of the knockdown levels.

12, Suppl. Fig 4B: the authors write: 'Western blotting confirmed ... the homozygous insertion of the targeting cassette into the STAG1 locus, ...' . I think the WB nicely shows insertion of the V5 tag into the STAG1 locus, but it I think WB cannot show homozygous insertion. The fact that in Suppl Fig 1B STAG1 expression is (almost) completely ablated, is a good indication, but in Fig. 2J, there is still about 50% expression. As such, proofing homozygous insertion by PCR/Sanger sequencing or densitometry over several experiments or just rephrasing the text a little might be beneficial.

We agree with the reviewer and have adapted the respective passage in the main text.

**Competing interests statement:** A patent application related to the design and use of the ARTi system entitled ‘Methods and molecules for RNA interference (RNAi)’ has been submitted by T.H., M.H., J.Z. and R.N. to the European Patent Office (application EP21217407.2).